# Electrodeposition of Ti-Doped Hierarchically Mesoporous Silica Microspheres/Tungsten Oxide Nanocrystallines Hybrid Films and Their Electrochromic Performance

**DOI:** 10.3390/nano9121795

**Published:** 2019-12-17

**Authors:** Ya Song, Zhiyu Zhang, Lamei Yan, Ling Zhang, Simin Liu, Shaowen Xie, Lijian Xu, Jingjing Du

**Affiliations:** 1College of Packaging Materials and Engineering, Hunan University of Technology, Zhuzhou 412008, China; sy18798448247@163.com (Y.S.); zzhiyu12@126.com (Z.Z.); 13502431554@163.com (S.X.); 2School of Digital Media and Design, Hangzhou Dianzi University, Hangzhou 310018, China; ylm@hdu.edu.cn; 3Hunan Key Laboratory of Biomedical Nanomaterials and Devices, College of Life Sciences and Chemistry, Hunan University of Technology, Zhuzhou 412007, China; lingzhang645@126.com (L.Z.); 18890229883@163.com (S.L.); xlj235@163.com (L.X.); 4National & Local Joint Engineering Research Center of Advanced Packaging Materials Developing Technology, Hunan University of Technology, Zhuzhou 412008, China

**Keywords:** Ti-doped hierarchically mesoporous silica, electrochromic, electrodeposition, tungsten oxide, hybrid film

## Abstract

In this paper, a novel Ti-doped hierarchically mesoporous silica microspheres/tungsten oxide (THMS/WO_3_) hybrid film was prepared by simultaneous electrodeposition of Ti-doped hierarchically mesoporous silica microspheres (THMSs) and WO_3_ nanocrystallines onto the fluoride doped tin dioxide (FTO) coated glass substrate. It is demonstrated that the incorporation of THMSs resulted in the hybrid film with improved electrochromic property. Besides, the content of THMSs plays an important role on the electrochromic property of the hybrid film. An excellent electrochromic THMS/WO_3_ hybrid film with good optical modulation (52.00% at 700 nm), high coloration efficiency (88.84 cm^2^ C^−1^ at 700 nm), and superior cycling stability can be prepared by keeping the weight ratio of Na_2_WO_4_·2H_2_O (precursor of WO_3_):THMSs at 15:1. The outstanding electrochromic performances of the THMS/WO_3_ hybrid film were mainly attributed to the porous structure, which facilitates the charge-transfer, promotes the electrolyte infiltration and alleviates the expansion of the film during Li^+^ insertion. This kind of porous THMS/WO_3_ hybrid film is promising for a wide range of applications in smart homes, green buildings, airplanes, and automobiles.

## 1. Introduction

Electrochromic materials that can reversibly change their optical properties such as transmittance [1], reflectance [2] and absorption [3] at the stimulation of a small electric field, have received significant interests owing to their potential applications in energy-saving smart windows [4,5,6], non-emissive information displays [7,8] and antiglare automotive mirrors [9]. It is recognized that the mechanism of the dynamic which changed optical properties of the electrochromic materials is mainly attributed to the electrochemically induced oxidation-reduction reaction [10,11]. Generally, materials containing variable valence elements can be used as electrochromic materials, including conducting polymers [12,13,14], organic small molecular [15,16] and inorganic metal oxides [17,18,19,20].

Among all the electrochromic materials, tungsten oxide (WO_3_) as a popular-star electrochromic material has been broadly explored due to its high optical contrast and a relatively low production cost [21,22,23], as compared with other electrochromic materials. Besides, the WO_3_ electrochromic material can control the spectral response of electrochromic in the visible spectrum and the infrared (IR) region, which has great application in architecture, aerospace, information storage and artificial intelligence fields. It is generally accepted that the electrochromic phenomena of tungsten oxide are attributed to the injection/extraction of electrons and cations (H^+^, Li^+^), which promote the redox reaction of W^6+^ ↔ W^5+^, and resulted in a reversibly color change of transparent ↔ blue [24,25,26]. More et al. [27] have reported the preparation of WO_3_ electrochromic films by electrodeposition and the resultant WO_3_ film exhibited amorphous nature, closed packed nano-granular morphology, good coloration efficiency and visible high transmittance. However, unsatisfactory optical modulation [28], long switching time [29] and short life time limit [30] its electrochromic effectiveness. To overcome these obstacles, significant efforts have been made to synthesis of porous WO_3_ electrochromic films to enhance its electrochromic performance. For instance, Xie and coworkers [31] prepared the two-dimensional (2D) grid structured WO_3_ film using polystyrene nanofibers as sacrificial templates to achieve an enhancement of electrochromic performance. The enhanced electrochromic performance may be attributed to that of a porous structure facilitating the penetration of electrolytes onto the WO_3_ framework and shortening the ionic diffusion paths for lithium ions in porous films. Zhang et al. [32] presented a periodical bowl-like macroporous WO_3_ array film synthesized by using the assembled monolayer polystyrene spheres as a template. The resultant WO_3_ microbowl array film showed superior electrochromic property with coloration efficiency of 68 cm^2^ C^−1^, response times of 3.6 s (coloration) and 1.0 s (bleaching) in comparison with the dense film prepared without a PS template. Compared with the template method, simultaneous electrodeposition of WO_3_ and other nanomaterials for fabrication of composite WO_3_ film is more convenient and efficient in creation of a porous structure in WO_3_ film [33,34,35]. Song et al. [36] reported the fabrication of porous WO_3_ film by electrochemically polymerizing 3-chlorothiophene on the surface of WO_3_ film in the ionic liquid 1-butyl-3-metyllimidazolium hexafluorophosphate. Fu et al. [37] described the electrochemical deposition of amorphous WO_3_ nanocrystallines/electrochemical reduced graphene oxide (WO_3_/rGO) nanocomposite film. The electrochromic properties of the WO_3_/rGO film show significant improvement compared to WO_3_ films, which is mainly due to the synergistic incorporation of rGO into WO_3_. Cai et al. [38] have prepared TiO_2_@WO_3_ core/shell nanorod array composites by the combination of hydrothermal and electrodeposition method. The array films show remarkable enhancement of the electrochromic properties stemming from the core/shell structure and the porous space among the nanorod array.

Recently, there has been great interest in Ti-doped WO_3_ thin films because the addition of adequate dopant in WO_3_ not only modifies its structure but also improves the electrochromic performance. Acosta’s group [39] has employed a pulsed spray pyrolysis technique to fabricate Ti-WO_3_ thin films with improved electrochromic behavior. Karuppasamy [40] proved that the Ti-doped WO_3_ thin films prepared by co-sputtering titanium and tungsten in Ar + O_2_ atmosphere, which exhibited high optical modulation and coloration efficiency. Barawi et al. [41] presented a dual-band electrochromic device with high optical contrast based on the TiO_2_/V_2_O_5_//WO_3_ composite materials and the device was capable of selectively controlling the transmitted sunlight over the visible and the near-infrared regions. Cai et al. [42] introduced a template-free hydrothermal method for preparation of hierarchical structure Ti-doped WO_3_ thin films with enhanced electrochromic performance.

Herein, we reported the preparation of Ti-doped hierarchically mesoporous silica microspheres/tungsten oxide (THMS/WO_3_) hybrid film by simultaneous electrodeposition of Ti-doped hierarchically mesoporous silica microspheres (THMSs) and WO_3_ nanocrystallines onto the fluoride doped tin dioxide (FTO) coated glass substrate. The effect of the incorporation of THMSs in WO_3_ films on structure, morphology and electrochromic properties of WO_3_ films were investigated. Besides, the relationship between the structure and electrochromic properties including the electrochemical properties of the CV behavior, electric resistance and ion diffusion efficiency, and optical properties of transmittance modulation range, switching time, coloration efficiency and cycling stability of THMS/WO_3_ hybrid film was systematically evaluated.

## 2. Experimental Sections

### 2.1. Materials

Hexadecylpyridinium chloride (CPC), 4-nitrophenol, tetraethoxysilane (TEOS), ammonia solution (25 wt%), titanium oxysulfate hydrate (TiOSO_4_·nH_2_O) and hydrogen peroxide solution (H_2_O_2_, 30 wt%), were purchased from Aladdin (Aladdin, Beijing, China). Poly (acrylic acid) (PAA, *M*_w_ = 240,000 g/mol) in water (25 wt%), propylene carbonate (PC, 99.5%) was obtained from Acros (Acros, Shanghai, China). Sodium tungstate dihydrate (Na_2_WO_4_·2H_2_O, 99%) and lithium perchlorate (99%) was obtained from Innochem (Innochem, Beijing, China). The fluorine-doped tin oxide (FTO) glasses with a thickness of 2.2 mm were purchased from Yaoke Photoelectric Co., Ltd. (Yaoke Photoelectric Co., Ltd., Huaian, China) THMSs were synthesized by dynamic template method, employing the mesomorphous complexes of cationic surfactant CPC and anionic polyelectrolyte PAA as template, TiOSO_4_ as titanium source and TEOS as silica source, as discussed elsewhere [43,44,45]. HMSs were prepared with the similar method without adding of TiOSO_4_. All the chemical agents were used without further purification.

### 2.2. Fabrication of Porous THMS/WO_3_ Hybrid Films

Porous THMS/WO_3_ hybrid films were fabricated by electrochemical deposition performed on an CHI440C electrochemical workstation at room temperature using a three-electrode system, where a Pt sheet (2 cm^2^) was used as the counter electrode and Ag/AgCl (in 3 M KCl aqueous solution) as the reference electrode, and FTO-coated glass substrate as work electrode. Initially, the FTO were sequentially sonicated in DI water, 0.1 M HCl solution and ethanol for 15 min, respectively, and then dried with N_2_ before usage. The peroxo tungstic acid (PTA) solution was prepared by dissolving 6 mmol (1.98 g) Na_2_WO_4_·2H_2_O in 100 mL deionized water with stirring at room temperature for 2 h, followed by adding of 2.8 mL H_2_O_2_ and 1.2 mL HNO_3_. After that, certain amounts (0.10 g–0.40 g) of THMSs were added into PTA solution under stirring. The THMS/WO_3_ hybrid films were electrodeposited onto FTO-coated glass substrate from the precursor solution at a constant potential of −0.47 V for 25 min. The as-deposited THMS/WO_3_ hybrid film was rinsed with DI water and then dried in air at room temperature for 2 h. By keeping the weight ratio of Na_2_WO_4_·2H_2_O:THMSs at 20:1, 15:1, 10:1 and 5:1, four THMS/WO_3_ hybrid films were prepared. The hybrid films were named as THMS/20WO_3_, THMS/15WO_3_, THMS/10WO_3_, and THMS/5WO_3_, respectively. The pure WO_3_ film was prepared by the similar method without adding of THMSs. Another reference hybrid film of HMSs/15WO_3_ was also prepared by electrochemical deposition by the adding of HMSs in the PTA solution, and the weight ratio of Na_2_WO_4_·2H_2_O:HMSs was kept at 15:1.

### 2.3. Electrochemical Characterization

The cyclic voltammetry (CV) and chronoamperometry (CA) measurements were performed in a three-compartment system containing 1 M lithium perchlorate/propylene carbonate (LiClO_4_/PC) as electrolyte, Ag/AgCl as a reference electrode and Pt foil as the counter electrode. The voltage supply was from CHI440C electrochemical workstation. Electrochemical impedance spectroscopy (EIS) tests were conducted on this electrochemical workstation with a superimposed 5 mV sinusoidal voltage in the frequency range of 0.05 Hz–100 kHz. The spectro-electrochemical properties of the films were measured using a CHI440C (CH Instruments, Inc., Austin, Texas, USA) and a UV-Vis-NIR spectrophotometer (UV-3600PLUS220/230VC, Shimadzu, Kyoto, Japan). The transmission spectra of the films were recorded in the wavelength range of 300–1600 nm at potentials of −1 V and 1 V. The situ transmission spectra (700 nm) of films under continuous double potentiostatic measurements between −1 V and 1 V with at time interval of 40 s were measured by combination of a Shimadzu UV-3600PLUS220/230VC spectrophotometer and CHI440C electrochemical workstation. The coloration efficiencies of the electrochromic films were measured according to the methods reported by Li et al. [46].

### 2.4. Sample Characterization

Scanning electron microscopy (SEM) images were obtained with TESCAN MIRA3 LMU instrument (TESCAN, Brno, Czech Republic). X-ray diffraction (XRD) using a Rigaku Model D/max-2500 (Rigaku Corporation, Tokyo, Japan) diffractometer, with Cu Kα radiation in the 2θ range of 10–80° with a step size of 0.02°. Transmission electron microscopy (TEM) observations and elemental analysis for Si/Ti ratios were performed on a JEM-1011 (JEOL, Tokyo, Japan) electron microscope, working at 100 kV, whereby a small drop of the sample was deposited onto a carbon-coating copper grid and dried at room temperature under atmospheric pressure. X-ray photoelectron spectroscopy (XPS) spectra were obtained with a Thermo Fisher ESCALAB 250Xi (Thermo Fisher Scientific, Waltham, MA, USA) system, using a monochromatic Al Kα X-ray source.

## 3. Results and Discussion

The THMS/WO_3_ hybrid films were prepared by a simple in situ electrodeposition process. Initially, THMSs were synthesized by the “dynamic template” method employing the mesomorphous complex of CPC and PAA as template, TEOS as silica source and TiOSO_4_ as titanium source. Subsequently, the deposition of THMS/WO_3_ hybrid films onto the FTO-coated glass substrate were achieved by electrochemical deposition of the precursor solution containing Na_2_WO_4_·2H_2_O and THMSs in a three-electrode system, where a Pt sheet as the counter electrode, Ag/AgCl as the reference electrode, and FTO-coated glass substrate as work electrode. Finally, the electrochromic properties of the THMS/WO_3_ hybrid films were studied by stepping the voltages between −1 V (deep blue) and 1 V (transparent). The whole preparation procedure was schematically shown in Scheme 1.

### 3.1. Synthesis of Ti-Doped Hierarchically Mesoporous Silica Microspheres (THMSs)

Figure 1 showed the SEM, TEM and the TEM elemental mapping of the synthesized THMSs. As shown in the SEM image (Figure 1A), the synthesized THMSs was uniform with average diameter of ~500 nm. From the SEM image at high magnification (the inset of Figure 1A), it was found that the surface of the THMSs was rough and contained obvious pores. TEM images gave a more deep insight of the porous structure of the THMSs. As shown in Figure 1B, hierarchically porous structure with plenty of mesopores and interstitial pores throughout the entire particle of the THMSs were clearly observed. The high resolution TEM image (Figure 1C) showed that the mesopores with pore size of approximately 2–3 nm in THMSs were in highly periodic order. Besides, the interstitial pores were well co-existed in THMSs, without disturbing the ordering of the mesopores. The elemental composition of the THMSs was checked by TEM elemental mapping. As shown in Figure 1D–F, the elements including O, Si, Ti were detected and found to be homogenously distributed in THMSs. The elemental content of Ti was measured to be 2.56% by EDXS (Appendix A).

### 3.2. Fabrication of THMS/WO_3_ Hybrid Films

The THMS/WO_3_ hybrid films were prepared by simultaneous electrodeposition of the THMSs and WO_3_ onto the FTO-coated glass substrate. The ratio of Na_2_WO_4_·2H_2_O and THMSs had a great impact on the morphology of the synthesized THMS/WO_3_ hybrid films. Here, we take the THMS/15WO_3_ sample prepared with the weight ratio of Na_2_WO_4_·2H_2_O:THMSs at 15:1, as a typical example to study the physical properties of the synthesized THMS/WO_3_ hybrid films. Figure 2 shows the XRD patterns of the FTO-coated glass substrate (FTO film), pure WO_3_ deposit onto the FTO-coated glass substrate (WO_3_ film), and THMS/15WO_3_ film, with the standard cubic phase SnO_2_ as a reference. As shown, the XRD pattern of the FTO film was resembled as that of the standard pattern of cubic phase SnO_2_ (JCPDS No.41-1445). For comparison, we prepared the pure WO_3_ film by electrodeposition of pure WO_3_ on the FTO-coated glass substrate. The XRD pattern of pure WO_3_ film displayed not only the sharp peaks corresponded to the cubic phase SnO_2_, but also an additional broadened peak around 2θ ≈ 26.62°. The additional peak was corresponded to the amorphous WO_3_. The XRD pattern of THMS/15WO_3_ film was similar to that of the WO_3_ film, indicating that the incorporation of THMSs did not significantly change the amorphous nature of the WO_3_. In general, nanocrystallines WO_3_ with amorphous background are suitable for electrochromic applications since it is favorable for ions to diffuse through.

The morphology of the as-prepared WO_3_ and THMS/15WO_3_ hybrid films were further compared by SEM observation. As shown, the pure WO_3_ film was compact and composed of WO_3_ nanocrystallines (Figure 3A,B). Differently, the THMS/15WO_3_ film was much rougher with obviously porous structure, as shown in Figure 3C. The THMS/15WO_3_ film was composed of WO_3_ nanocrystallines and THMSs as indicated by the blue stained part. The magnified SEM image (Figure 3D) further demonstrated the existence of the THMSs, indicating the successful fabrication of THMS/WO_3_ hybrid film.

X-ray photoelectron spectroscopy (XPS) measurements [47] were performed to analyze the surface elemental composition and the valence state of the main elements in the THMS/WO_3_ films. The XPS survey spectrum of THMS/15WO_3_ film showed that the hybrid film contains C, W, O, elements since the peaks at specific binding energy of 285.00 eV, 36.12 eV, and 531.00 eV were successfully detected, as shown in Figure 4a. The high-resolution W 4f core-level XPS spectrum was shown in Figure 4b. The spin-orbit doublets in this spectrum corresponding to W (VI) 4f_7/2_, W (VI) 4f_5/2_ and W (VI) 5p_3/2_ peaks located at 36.19 eV, 38.30 eV and 42.20 eV were clearly observed, indicating that W was at its highest oxidation state (W^6+^) [48]. The Ti 2p and Si 2p peaks were not well resolved due to the low content of THMSs in THMS/15WO_3_ film. However, it can be clearly resolved in the high-resolution XPS spectrum. The existence of Ti element in THMS/15WO_3_ film was demonstrated by the high-resolution XPS spectrum at the range of 469 eV to 453 eV, as shown in Figure 4c. Besides, the Ti 2p_3/2_ and Ti 2p_1/2_ peaks were located at 458.6 eV and 464.3 eV in the Ti 2p spectrum, verifying that the Ti in the THMS/15WO_3_ film was in the highest oxidation state (Ti^4+^). Moreover, the high-resolution XPS spectrum at the range of 111 eV to 96 eV showed an obvious Si 2p (Figure 4d) peak at 104.18 eV, demonstrating that the THMS/15WO_3_ film contains Si element.

By varying the weight ratio of Na_2_WO_4_·2H_2_O:THMSs in the precursor solution, we prepared a series of THMS/WO_3_ hybrid films with the weight ratio of Na_2_WO_4_·2H_2_O:THMSs at 20:1, 15:1, 10:1, and 5:1, respectively. The resultant THMS/WO_3_ hybrid films were denoted as THMS/20WO_3_, THMS/15WO_3_, THMS/10WO_3_ and THMS/5WO_3_, respectively. The morphology of the resultant THMS/WO_3_ hybrid films were observed by SEM. As shown in Figure 5, THMS/20WO_3_ (Figure 5A,B) and THMS/10WO_3_ (Figure 5C,D) hybrid films exhibited a rough surface with obviously porous structure and both the WO_3_ nanocrystallines and THMSs were clearly seen (the blue stained part). The morphology is similar to that of the THMS/15WO_3_ hybrid films (Figure 3C,D). Differently, the amount of the deposited THMSs on the THMS/WO_3_ hybrid films increased with decreasing the weight ratio of Na_2_WO_4_·2H_2_O:THMSs. The SEM images of the THMS/5WO_3_ hybrid films were shown in Figure 5E,F. As shown, very few WO_3_ nanocrystallines and THMSs were deposited on the hybrid film. This may be due to the poor electrical conductivity of the THMSs than that of WO_3_ nanocrystallines. Too many THMSs in the precursor solution prevented the deposition of WO_3_ nanocrystallines and THMSs onto the FTO glass substrate.

### 3.3. Electrochromic Properties of THMS/WO_3_ Hybrid Films

#### 3.3.1. Electrochemical Performances

In order to check the effect of THMSs incorporation on the electrochemical performances of the WO_3_ film, the CV behavior of the WO_3_ and THMS/WO_3_ hybrid films with different THMSs content were comparatively studied in a standard three-electrode system. The CV curves of the five films were measured in the potential region of −1 V~1 V at a scan rate of 50 mV s^−1^. The CV curves of the five films exhibited some observable similarities and differences, as shown in Figure 6a. At first glance, the shapes of the CV curves were similar for all the films. Nevertheless, the enclosed area of the CV curves was different from each other. Generally, the area of the CV curves indicates the amount of charges inserted or extracted from the samples, which corresponded to the electrochemical performance of the materials. After careful observation, we found that the enclosed area of the CV curves was correlated with the content of THMSs. With the decreasing of the weight ratio of Na_2_WO_4_·2H_2_O:THMSs, which means increasing the content of THMSs in composite film, the area of the CV curves was initially increased with the weight ratio of Na_2_WO_4_·2H_2_O:THMSs beyond 15:1 and then decreased. Clearly, the CV curves of the THMS/15WO_3_ film exhibited the largest enclose area. This may be due to that the THMS/15WO_3_ film had the largest surface area as indicated by the obviously porous surface structure. The large surface area facilitated the insertion/extraction of small ions (Li^+^) into the host lattice. Besides, the doping of Ti^4+^ can increase the conductivity of the hybrid film. This can be demonstrated by the comparison of the CV curves of the THMS/15WO_3_ and HMS/15WO_3_ films, as shown in Appendix A. Moreover, the CV curves of the THMS/15WO_3_ exhibited the larger cathode and anode peak current densities, which reflects the fact that proton insertion/extraction into the host lattice is facilitated at a given applied potential. Furthermore, the onset potential of the cathodic current for the THMS/15WO_3_ film is strongly shifted in the positive direction compared to the pure WO_3_ film; that is, insertion can be achieved at a considerably low applied voltage. The reason why the THMS/15WO_3_ film showed superior electrochemical performance is discussed. As we know, the electrochemical performance of the electrochromic film is highly depended on the structure and composition. The incorporation of THMSs in WO_3_ film could endow the composite film with porous structure, which offer an easy path to charge transfer and diffusion processes of ions. For example, when the weight ratio of Na_2_WO_4_·2H_2_O:THMSs was kept beyond 10:1, the prepared THMS/WO_3_ composite films presented obviously porous structure (Figure 5A,C and Figure 3C). On the other hand, the incorporation of THMSs will hinder the oxidation-reduction of PTA on the surface of the FTO substrate and decrease the electroactive area of the composite film. This is quite obvious when the weight ratio of Na_2_WO_4_·2H_2_O:THMSs was kept at 5:1. The resultant THMS/WO_3_ composite film presented as compact WO_3_ film, as shown in Figure 5E. Thus, there exists optimal weight ratio of Na_2_WO_4_·2H_2_O:THMSs of 15:1 to achieve THMS/WO_3_ composite film with the best electrochemical performance from a trade-off between these two contradictory effects.

The electric resistance is another important parameter to assess the electrochemical performance of electrochromic materials [49]. Thus, we further compared the resistance of the WO_3_ and THMS/WO_3_ film electrodes by EIS measurements, which were conducted by applying an AC voltage of 5 mV in a frequency range of 0.05 Hz to 100 kHz at their bleached state (about 0.5 V vs. Ag/AgCl). Figure 6b showed the Nyquist plots of WO_3_ and THMS/WO_3_ hybrid films. As shown, all the plots contained a semicircle at high-frequency represented the charge-transfer impedance on the electrode/electrolyte interface, and a straight line in the low frequency region correlated with ion diffusion process within the electrode. The THMS/15WO_3_ hybrid films exhibited the smallest semicircle suggesting the lowest charge transfer resistance, and a straight line close to the theoretical vertical line implying the high ion diffusion rate.

To get a quantitative comparison of the ion diffusion efficiency in the WO_3_ and THMS/WO_3_ film electrodes, the CV curves of the five films at different scan rates ranging from 10 to 100 mVs^−1^ were measured. The CV curves are shown in Appendix A. The peak current *I_p_* (Amperes), during anodic scans at different scanning rates was used to extract the diffusion coefficient *D* (cm^2^ s^−1^) of Li^+^ in the electrode, which could be calculated by employing the Randles–Sevcik equation (Equation (1)):
*I_p_* = 2.71 × 10^5^*ACn*^3/2^*D*^1/2^*v*^1/2^(1)
where *D* was the diffusion coefficient of Li^+^ ions, *v* was the scan rate, *n* was the number of electrons and it was assumed to be 1, *A* was the interface between the electrolyte and the active material, *C* was the concentration of active ions in the electrolyte solution. The calculated diffusion coefficients of the five films were summarized in Table 1. As shown, the THMS/15WO_3_ hybrid film exhibited the highest *D* value (1.16 × 10^−6^ cm^2^ s^−1^), which was about 1.3 times higher than that of the pure WO_3_ (8.82 × 10^−7^ cm^2^ s^−1^).

#### 3.3.2. Optical Performances

The optical properties of WO_3_ and THMS/WO_3_ hybrid films were measured after the film electrodes had been subjected to CV testing for 10 cycles in 1 M LiClO_4_/PC. The transmittance of the samples at the colored and bleached states by applying step voltages of −1 V and 1 V (vs. Ag/AgCl) for 40 s, respectively, were recorded over a wavelength region from 300 to 1600 nm (Figure 7a). During cycling, the color of all the film changed reversibly from transparent to deep blue along with reversible surface redox reactions. The switching mechanism could be described by Equation (2):
WO_3_ + *y*Li^+^ + *y*e^−^ ↔ Li*_y_*WO_3_(2)
where *W*(V) was in blue color and *W*(VI) was transparent. As shown in Figure 7a, all the hybrid films exhibited the large transmittance modulation range (transmittance contrast in bleached and colored states) at both VIS and NIR regions. The transmittance spectra of the samples at open circuit potential were shown in Figure 7b. At open circuit, potential of prepared WO_3_ and THMS/WO_3_ hybrid films were in blue color with low transmittance of VIS and NIR radiations. We chose the wavelength at 700 nm as a typical example to make a quantitative study on the modulation ranges of the THMS/WO_3_ hybrid films. The modulation ranges (*ΔT*) of the transmittance (*ΔT* = *T_b_* − *T_c_*, where *T_b_* and *T_c_* denoted as transmittance in bleached and colored states, respectively) for the WO_3_ and THMS/WO_3_ hybrid films at 700 nm were given in Table 1. The modulation range of the transmittance could reach up to 32.49%, 48.77%, 52.00% and 37.77% at 700 nm for the THMS/5WO_3_, THMS/10WO_3_, THMS/15WO_3_ and THMS/20WO_3_ hybrid films, respectively. While the pure WO_3_ film only exhibited 34.10% at 700 nm. The THMS/15WO_3_ film showed the highest optical contrasts at 700 nm.

The switching time is another crucial parameter that reflects the electrochromic property of the materials [50]. The switching characteristics of the pure WO_3_ and THMS/WO_3_ hybrid films were investigated by CA and the corresponding in situ transmittance at 700 nm during continuous double potentiostatic measurements between −1 V and 1 V at a time interval of 40 s, as shown in Figure 8a,b. The values of *t_c_* and *t_b_* (*t_c_* and *t_b_* denotes as the colored time and the bleached time, respectively) for all the films were given in Table 1. As shown, for the THMSs incorporated THMS/WO_3_ hybrid films, the switching speed for full bleached and colored was faster than that of pure WO_3_ film. This may be attributed to the porous structure of the THMS/WO_3_ hybrid films. Synthetically, comparison showed that the THMS/15WO_3_ hybrid films exhibited the best switching characteristics with *t_c_* of 14.50 s and *t_b_* of 11.83 s. Based on the transmittance of the electrochromic film in bleached and colored states (Figure 8a,b), the coloration efficiency of electrochromic film, which is defined as the change in optical density (*ΔOD*) resulting from the charge density (*ΔQ*) inserted into (or extracted from) the electrochromic material, can be calculated according to Equation (3).
*CE* = *ΔOD*/*ΔQ* = log(*T_b_*/*T_c_*)/*ΔQ*(3)
where *T_b_* and *T_c_* refer to the transmittance of the electrochromic film in bleached and colored states, respectively. Figure 8c showed the plots of *ΔOD* at a wavelength of 700 nm vs. the inserted charge density for the oxides during the coloration process. The slope of the line fitting the linear region of the curve represented the coloration efficiency of the electrochromic films. The coloration efficiency value of pure WO_3_ film was calculated to be 46.84 cm^2^ C^−1^ at 700 nm, while the coloration efficiency value for THMS/5WO_3_, THMS/10WO_3_, THMS/15WO_3_, and THMS/20WO_3_ were 41.17, 68.86, 88.84 and 58.62 cm^2^ C^−1^, respectively. Clearly, the THMS/15WO_3_ hybrid film possessed the highest coloration efficiency.

The above discussions have demonstrated that the THMS/15WO_3_ hybrid film exhibited the most excellent electrochromic properties. Finally, the electrochromic stability of the THMS/15WO_3_ hybrid film was checked by the comparison of the digital photo of the hybrid film at colored and bleached state before and after 100 times of redox cycles. The digital photos are shown in Figure 8d. As shown, the THMS/15WO_3_ hybrid film still displayed a deep blue color at colored state and nearly transparent at the bleached state after 100 times of redox cycles, indicating the good electrochromic stability of the THMS/15WO_3_ hybrid film. Therefore, we obtained an excellent electrochromic THMS/WO_3_ hybrid film with good optical modulation, high coloration efficiency, and excellent cycling stability by keeping the weight ratio of Na_2_WO_4_·2H_2_O (precursor of WO_3_):THMSs at 15:1. These properties endowed them to be effective candidates in various applications, such as large area information displays, rear-view mirrors for automobiles, thermal control of spacecraft and military camouflage.

## 4. Conclusions

In summary, we reported the successful fabrication of THMS/WO_3_ hybrid film by electrochemical deposition method. The incorporation of THMSs in WO_3_ film resulted in the hybrid film with porous surface structure, which significantly increase the surface area of the hybrid film. It is demonstrated that the content of THMSs in the THMS/WO_3_ hybrid film plays an important role on the morphology and electrochromic property of the hybrid film. We obtained an excellent electrochromic THMS/WO_3_ hybrid film with good optical modulation (52.00% at 700 nm), high coloration efficiency (88.84 cm^2^ C^−1^ at 700 nm), and excellent cycling stability by keeping the weight ratio of Na_2_WO_4_·2H_2_O (precursor of WO_3_):THMSs at 15:1. The outstanding electrochromic performances of the porous THMS/WO_3_ hybrid film were mainly attributed to the porous surface structure, and proper amount of Ti-doping improved the electric conductivity of the hybrid film, which facilitates the charge-transfer, promotes the electrolyte infiltration and alleviates the expansion of the film during Li^+^ insertion. We envision that the as-prepared THMS/WO_3_ hybrid film will have great potential application in architecture, aerospace, information storage and artificial intelligence fields.

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
