# Peer review of "Electrodeposition of Ti-Doped Hierarchically Mesoporous Silica Microspheres/Tungsten Oxide Nanocrystallines Hybrid Films and Their Electrochromic Performance"

_nanomaterials, 2019, doi:10.3390/nano9121795_

Round 1

Reviewer 1 Report

The authors present an investigation of the electrochromic advantages Mesoporous Silica Microspheres/Tungsten oxide Nanocrystallines Hybrid Films if compared with Tungsten oxide. 

The work is in general well organised and the authors present an interesting hybrid materials which can improve the WO3 electrochromic behavior.

The introduction is clear and put in the context of the thematic, although the mention of new dual band devices that are capable of selectively filtering NIR and VIS or used both electros as active electrochromic materials is missed. Author shoud mention that and cite some related works, as for example J. Mater. Chem. A, 2018,6, 10201-10205, in wich WO3 is used as electrode with V2O5 to achieve selectivity solar filtration. The preparation of the materials and the structural and morphological characterization is well described and explained. 

However, in the electrochemical characterization there are many gaps that must be resolved. First of all they said that they are doing Electrochemical impedance spectrum whereas the rigth name of thuis tecnique is Electrochemical impedance spectroscopy. The have to chenge that. 

Related with the CV, the author claims that "Clearly, The CV curves of the THMS/15WO3 film exhibited the largest enclose area implying that the content of THMSs in the WO3 films had great impact on the electrochemical performance of the hybrid film" but the trend of the curves does not make to much sense. THMS/5WO3 present a lower area, whereas THMS/10WO3 and THMS/15WO3 higer, can the author explaine why? They also claims that "the onset potential of the cathodic 262 current for the THMS/5WO3 and THMS/20WO3 hybrid films was negatively shifted. This meant that the ion insertion in THMS/10WO3 and THMS/15WO3 hybrid films could be achieved at a considerably low applied voltage" and this is not clear lokking into the CV measures. 

Looking into the optical performances, first of all, an analysis of the behaviour of the materials in the NIR is missing. WO3 it is a well known electrochromic material able to filter the NIR, and the investigation of these hybris materials in this part of the spectrum shoud be performed. This part of the analysis it is really not clear for me.  The transmittance spectra of the samples (Figure 7a) present the transmitance at +1 and -1V. Why the auhor not present the OCP transmitance situation? This shoud be the firt measure. 

The trend of the measures it is again kind of confusing, and does not make sense. Why again here THMS/5WO3 present worst electrochromic performance than WO3? Authors should explain that. 

The rest of the interpretations seems to be fine, but before publication, authors should elaborate a better explanation of the electrochemical and  optical performance part, and likewise repeat the transmittance experiments to investigate what happens in the NIR part of the spectrum.

Author Response

Responses to Reviewers

Title: Electrodeposition of Ti-doped Hierarchically Mesoporous Silica Microspheres/Tungsten oxide Nanocrystallines Hybrid Films and their Electrochromic Performance

Manuscript ID: nanomaterials-646760

Dear Editor Zoe Li

We are grateful to you for promptly handling the submission of our manuscript and for finding knowledgeable experts to review our manuscript. We sincerely thank the reviewers for their insightful comments, which have helped us to improve the quality of the manuscript. We have performed several additional experiments and prepared a point-by-point response to the reviewer comments. Changes in the text and responses to the reviewer comments are marked in red. I think the revised manuscript is more polished, which would be interesting for the reader. Thank you very much for considering our manuscript for publication after revision in your prestigious journal Nanomaterials.

Thank you very much for your consideration. 

Best regards!

Yours sincerely

Jingjing Du

Reviewer #1

Comment

The authors present an investigation of the electrochromic advantages Mesoporous Silica Microspheres/Tungsten oxide Nanocrystallines Hybrid Films if compared with Tungsten oxide. The work is in general well organised and the authors present an interesting hybrid materials which can improve the WO3 electrochromic behavior.

Response: Thank you so much for your encouragement and positive comments.

Comment

The introduction is clear and put in the context of the thematic, although the mention of new dual band devices that are capable of selectively filtering NIR and VIS or used both electros as active electrochromic materials is missed. Author shoud mention that and cite some related works, as for example J. Mater. Chem. A, 2018,6, 10201-10205, in which WO3 is used as electrode with V2O5 to achieve selectivity solar filtration. The preparation of the materials and the structural and morphological characterization is well described and explained.

Response: We greatly appreciate that the reviewer gave us a helpful suggestion. We have introduced electrochromic materials of WO3 can control the spectral response of electrochromic in the visible spectrum and the infrared (IR) region. Meanwhile, some related references were cited in the revised manuscript.

Comment

However, in the electrochemical characterization there are many gaps that must be resolved. First of all they said that they are doing Electrochemical impedance spectrum whereas the rigth name of this technique is Electrochemical impedance spectroscopy. The have to change explained.

Response. Thank you for your careful reading of our manuscript. The expression of Electrochemical impedance spectrum has been changed to Electrochemical impedance spectroscopy in the revised manuscript.

Comment

Related with the CV, the author claims that "Clearly, The CV curves of the THMS/15WO3 film exhibited the largest enclose area implying that the content of THMSs in the WO3 films had great impact on the electrochemical performance of the hybrid film" but the trend of the curves does not make to much sense. THMS/5WO3 present a lower area, whereas THMS/10WO3 and THMS/15WO3 higher, can the author explaine why? They also claims that "the onset potential of the cathodic 262 current for the THMS/5WO3 and THMS/20WO3 hybrid films was negatively shifted. This meant that the ion insertion in THMS/10WO3 and THMS/15WO3 hybrid films could be achieved at a considerably low applied voltage" and this is not clear looking into the CV measures.

Response: Thank you so much for your comments. The description and discussion of the CV curves have been rewritten (Page 9, 263-277). The reason why the THMS/15WO3 film showed superior electrochemical performance is discussed (Page 9, line 277-289). As we know, the electrochemical performance of the electrochromic film is highly depended on the structure and composition. The incorporation of THMSs in WO3 film could endow the composite film with porous structure, which offer an easy path to diffusion and charge transfer processes of ions. For example, when the weight ratio of Na2WO4•2H2O : THMSs was kept beyond 10 : 1, the prepared THMS/WO3 composite films presented obviously porous structure (Fig. 5A, Fig. 3C and Fig. 5B). On the other hand, the incorporation of THMSs will hinder the oxidation-reduction of Na2WO4 on the surface of the FTO substrate and decrease the conductivity of the composite film. This is quite obvious when the weight ratio of Na2WO4•2H2O : THMSs was kept at 5 : 1. The resultant THMS/WO3 composite film presented as compact WO3 film as shown in Fig. 5C. Thus, there exists optimal weight ratio of Na2WO4•2H2O : THMSs of 15 : 1 to achieve THMS/WO3 composite film with the best electrochemical performance from a trade-off between these two contradictory effects.

Comment

Looking into the optical performances, first of all, an analysis of the behaviour of the materials in the NIR is missing. WO3 it is a well known electrochromic material able to filter the NIR, and the investigation of these hybris materials in this part of the spectrum shoud be performed. This part of the analysis it is really not clear for me. The transmittance spectra of the samples (Figure 7a) present the transmitance at +1 and -1V. Why the auhor not present the OCP transmitance situation? This shoud be the firt measure.

Response: Thank you for your instructive suggestions. The optical transmittance spectra of the WO3 and THMS/WO3 hybrid films in the wavelength range of 300 - 1600 nm under applying alternate potentials of -1 V and 1 V and the optical transmittance spectra of the WO3 and THMS/WO3 hybrid films at open circuit potential have been added in the revised manuscript (Page 11, Figure 7). The description of Figure 7 has been added in the revised manuscript (Page 10).

Comment

The trend of the measures it is again kind of confusing, and does not make sense. Why again here THMS/5WO3 present worst electrochromic performance than WO3? Authors should explain that.

Response. Thank you for your insightful suggestions. The reason why the THMS/15WO3 film showed supior electrochemical performance and THMS/5WO3 present worst electrochromic performance than WO3 are discussed (Page 9, line 277-289).

Comment: The rest of the interpretations seems to be fine, but before publication, authors should elaborate a better explanation of the electrochemical and optical performance part, and likewise repeat the transmittance experiments to investigate what happens in the NIR part of the spectrum.

Response: Thank you so much for your encouragement and positive comments. The detailed information for experimental replication has been added in the revised manuscript.

Reviewer #2

Comment: In this paper authors have investigated the quality of Ti-doped hierarchically mesoporous silica microspheres/tungsten oxide (THMS/WO3) hybrid film by simultaneous electrodeposition of THMSs and WO3 nanocrystallines onto the fluoride doped tin dioxide (FTO) coated glass substrate. The relationship between the film structure and electrochromic properties including the electrochemical properties of the CV behavior, electric resistance and ion diffusion efficiency, and optical properties of transmittance modulation range, switching time, coloration efficiency and cycling stability of THMS/WO3 hybrid films were systematically investigated. Authors have found an excellent electrochromic THMS/WO3 hybrid film behaviours with good optical modulation, high coloration efficiency and excellent cycling stability by keeping the weight ratio of Na2WO4·2H2O (precursor of WO3) : THMSs at 15:1. The excellent electrochromic performances of the THMS/WO3 hybrid film were mainly due to the porous structure, which facilitates the charge-transfer, promotes the electrolyte infiltration and alleviates the expansion of the film. This kind of porous THMS/WO3 hybrid films are promising for a wide range of applications in smart homes, green buildings, airplanes, and automobiles. This manuscript is well written by analysing the experimental data. It is important to publish this paper for the future technology. I recommend publishing this manuscript.

Response: Thank you so much for your encouragement.

Reviewer 2 Report

   Please, see the enclosed comments.

Author Response

(The authors gave the same response as above.)

Round 2

Reviewer 1 Report

The authors improved the electrochemical discussion and made the suggested corrections. Authors should correct the reference in line 82 and put the surname and not the name of the author.